# Using QTL to Identify Genes and Pathways Underlying the Regulation and Production of Milk Components in Cattle

**DOI:** 10.3390/ani13050911

**Published:** 2023-03-02

**Authors:** Thomas John Lopdell

**Affiliations:** Livestock Improvement Corporation, Hamilton 3240, New Zealand; thomas.lopdell@lic.co.nz

**Keywords:** mammary biology, mammogenesis, lactation, lactogenesis, quantitative trait loci

## Abstract

**Simple Summary:**

Milk and other dairy products are commonly consumed in many parts of the world. Dairy cattle, having millions of milk trait records, make excellent model species for understanding the genetics controlling the production of milk. This manuscript gives a summary of the current understanding of the genetic signals for milk production, in terms of the biological pathways they are involved with, and highlights a number of methods that can be used to identify the genes and variants underlying these signals. Knowledge of these variants will improve the ability of farmers and animal breeding companies to increase the rate of genetic gain for milk traits and enable the use of technologies such as gene editing.

**Abstract:**

Milk is a complex liquid, and the concentrations of many of its components are under genetic control. Many genes and pathways are known to regulate milk composition, and the purpose of this review is to highlight how the discoveries of quantitative trait loci (QTL) for milk phenotypes can elucidate these pathways. The main body of this review focuses primarily on QTL discovered in cattle (*Bos taurus*) as a model species for the biology of lactation, and there are occasional references to sheep genetics. The following section describes a range of techniques that can be used to help identify the causative genes underlying QTL when the underlying mechanism involves the regulation of gene expression. As genotype and phenotype databases continue to grow and diversify, new QTL will continue to be discovered, and although proving the causality of underlying genes and variants remains difficult, these new data sets will further enhance our understanding of the biology of lactation.

## 1. Introduction

The composition of milk is complex, featuring an emulsion of fat globules and a colloidal dispersion of casein micelles in an aqueous solution of lactose (and other carbohydrates), whey proteins, and minerals. Although milk from different species contains the same basic constituents, their proportions can vary greatly. In cattle, the typically average percentages (g/100 g) of fat, caseins, whey proteins, and lactose are 3.9%, 2.6%, 0.6%, and 4.6% respectively; in humans, the corresponding percentages are 4.5%, 0.4%, 0.5%, and 7.1% [1]. Even more extreme differences can be seen in other species. Some seal species, for example, producing little to no lactose, resulting in highly concentrated milk with fat percentages of 50% [2].

Less-extreme differences in milk composition are also visible within species. In many cases, the differences in composition among individuals are under partial genetic control. Regions of the genome where the genotypes of genetic variants are associated with phenotypes such as milk composition are known as quantitative trait loci (QTL). When they can be identified, the causative genes underlying these QTL can help elucidate the pathways involved in milk production. The aims of this review were to describe some of the major pathways required for milk production in terms of the QTL and genes that have helped to identify them and to note some of the methods that can be used to identify causative genes underlying QTL. We focus primarily on cattle (*Bos taurus*) as a model species, though there are some references to QTL identified in sheep; additionally, some comparisons with human milk composition are also presented where there is a major difference between the two species.

## 2. QTL for Major Pathways Involved in Milk Production

Many QTL have been observed for both milk yield and milk composition traits in cattle (see Table 1). These QTL have been identified using a number of different techniques, including linkage-based approaches, such as the transmission disequilibrium test (TDT); and association-based approaches, such as genome-wide association studies (GWAS) and transcriptome-wide association scans (TWAS). The genes attributed to these QTL have a variety of functions. Some, such as the hormones prolactin and growth hormone, and their associated signalling pathways, are involved in mammogenesis (the development of the mammary gland during puberty and pregnancy), lactogenesis (the onset of milk secretion), and galactopoiesis (the continued production of milk). Other pathways, such as those for fat and protein synthesis, and ion channels, are involved directly in milk production. The following sections list some of the pathways involved in mammogenesis and lactation, as identified by QTL for milk production, where the candidate causal gene encodes a protein that sits within those pathways (see summary in Figure 1). Methods for identifying candidate causative genes underlying QTL are discussed in Section 3.

### 2.1. Milk Proteins

QTL have been mapped to many milk proteins, i.e., those expressed directly in milk. In cattle, the largest proportion of milk protein content (80% [4]) consists of the four casein proteins, encoded by a cluster of genes mapping to BTA6: casein alpha-S1 (*CSN1S1*), encoding $\alpha_{S1}$-casein, casein alpha-S2 (*CSN1S2*) for $\alpha_{S2}$-casein, casein beta (*CSN2*) for β-casein, and casein kappa (*CSN3*) for κ-casein. The casein proteins aggregate into micelles in the milk, sequestering calcium phosphate as a nutrient source for the neonate. K-casein seems to be particularly important for successful lactation, as knockout mice deficient in κ-casein fail to lactate, due to destabilisation of the casein micelles [5]. Like β-lactoglobulin, the various casein proteins all have a range of coding variants, although, with the exception of *CSN1S1*G* [6], these have not been intrinsically linked to lower rates of gene expression, such as the β-lactoglobulin B variants discussed below. However, there is evidence that the SNP responsible for the A2 variant of β-casein is associated with both milk yield and protein yield [7], and QTL for milk protein phenotypes have also been detected at this locus in other studies [8,9]. These QTL can overlap with, but are not in linkage disequilibrium with, other nearby QTL assigned to the *GC* gene [8,10,11]: this gene encodes the protein group-specific component (vitamin D binding) and maps around one megabase from the casein gene cluster. Outside the mammary gland, casein expression in human CD14^+^ monocytes has been observed to upregulate expression of the cytokines granulocyte-macrophage colony-stimulating factor (GM-CSF), interleukin-1β (IL-1β), and IL-6 via the p38–MAPK pathway [12,13]. This suggests a possible role for caseins in regulating the innate immune response in the neonate intestine following milk consumption.

The major whey protein in cattle, accounting for around 50% of whey protein [4] and 10% of total protein, is β-lactoglobulin, which is encoded by the progestagen-associated endometrial protein (*PAEP*) on BTA11. A QTL has been detected to map to this gene. It has pleiotropic effects on milk fat yield, protein yield, and volume [8,14]. Other studies looking at individual milk proteins have also shown that genetic variants associated with low β-lactoglobulin concentrations yield high concentrations of α-, β-, and κ-caseins [15]. Variation in milk β-lactoglobulin concentrations are primarily driven by differences between the A and B protein variants [16,17]. The B variant showed lower expression than the A variant. Other variants mapping to the B variant background, such as B${}^{*}$ [18] and B${}^{\prime }$ [19], cause further reductions in β-lactoglobulin expression. Milk with low β-lactoglobulin concentrations has potential uses in infant formula (as human milk lacks this protein), and the associated higher concentrations of caseins would also be expected to provide better properties for cheese making.

The dominant whey protein in humans, which is second in cattle (18.5% of total whey protein [4]), is α-lactalbumin. It is encoded by the gene lactalbumin alpha (*LALBA*) on BTA5. The QTL associated with milk protein concentration have been identified at this locus in several dairy populations and breeds [20,21,22,23]. The α-lactalbumin protein is required for lactose synthesis, modifying the enzyme β-1,4-galactosyltransferase (encoded by *B4GALT1* on BTA8): in most tissues, B4GALT1 adds galactose moieties to N-acetylglucosamine (GlcNAc) residues. However, the binding of α-lactalbumin to B4GALT1 changes the latter to instead add galactose to glucose, forming lactose: this binary enzyme is known as lactose synthase. Given the importance of these two genes for lactose synthesis, it is no surprise that a QTL for milk lactose concentration has also been identified at the LALBA locus [24], along with a milk yield at the B4GALT1 locus [25]. The importance of lactose synthesis for milk production was shown in α-lactalbumin knockout mice, which produce highly viscous milk with otherwise normal fat and protein composition that cannot be extracted by the pups from the mammary gland [26]. It has been shown in vitro that multimeric α-lactalbumin is cytotoxic to stem cells and transformed cell lines, though it does not harm healthy epithelial cells [27], suggesting that α-lactalbumin may also have a protective function in either the mammary gland or the neotate digestive system.

Another whey protein is the iron-binding antimicrobial protein lactoferrin, encoded by the lactotransferrin (*LTF*) gene on BTA22. In cattle, this protein exists in bovine milk at low but variable concentrations (average 115.4 μg/mL in [28] but ranging from 31.8 to 485.6), but reaches much higher concentrations in human milk—around 2 g/L in mature milk [29]. Interestingly, another antimicrobial protein, lysozyme, which is present at lower levels in human milk [4], is also present at relatively low concentrations in cattle [30]. Levels of lysozyme in *Bovidae* are reported to be as low as 1/1000th of those of other mammalian species [31]. Although lactoferrin expression can be induced by mastitic infection [32], it is also under partial genetic control [33], and genetic variants affecting expression have been identified at the *LTF* locus [34]. QTL at this locus have also been associated with casein number and lactose concentration [35].

A third antibacterial protein present in milk is lactoperoxidase, encoded by the *LPO* gene on BTA19. In contrast to lysozyme and lactoferrin, lactoperoxidase activity was found to be around 20× higher in bovine milk compared to that of humans [36]. A *trans*-eQTL for *LPO*, overlapping a QTL for milk protein concentration, has been identified on BTA20 at the locus of the *C6* and *C7* genes [37]: these two genes encode proteins that comprise part of the complement pathway in the innate immune system, suggesting that lactoperoxidase may be co-regulated with this system.

### 2.2. Fat Synthesis Pathways

Fat is one of the major components of milk, forming membrane-bound droplets known as milk fat globules (MFG), mostly in the form of triglycerides. MFG membranes also contain a range of proteins, such as butyrophilin, adipophilin, mucin, lactadherin, lactoferrin, and xanthine oxidase [38]. Two of these, butyrophilin (encoded by butyrophilin subfamily 1 member A1; *BTN1A1*) and xanthine oxidase (*XOR*; encoded by xanthine dehydrogenase XDH), are required for enveloping the MFGs with the apical cell membrane, and therefore, for secretion of the MFG into milk [39]. Heterozygous knockout mice for *XOR* are unable to maintain lactation [40]. Many of the QTL for fat yield or milk concentration in cattle map to genes encoding fat synthesis or metabolic enzymes (see Figure 2).

One of the most prominent QTL detected in cattle [41] for milk volume, fat, and protein phenotypes in cattle maps to the *DGAT1* locus on BTA14 [42]. This gene encodes the enzyme diacylglycerol O-acyltransferase 1, which is responsible for the final stage triglyceride fat synthesis [43]. The causative variant for this QTL is a non-conservative lysine-to-alanine substitution at position 232 [42]. More recently, this same variant has been shown to cause an expression QTL for the *DGAT1* gene by disrupting an exon splice enhancer, which in turn alters the splicing efficiency of several introns in the transcript [44]. Another enzyme sitting earlier in the same trigyceride synthesis pathway is glycerol-3-phosphate acyltransferase 4, encoded by the *GPAT4* gene (formerly known as *AGPAT6*) on BTA27. Like *DGAT1*, a highly pleiotropic QTL has been detected at this locus for many milk phenotypes, including volume, fat, protein, and lactose traits [45]. The enzyme lipin 1 sits in a related pathway, catalysing the conversion of phosphatidate into diacylglycerol and functioning as a transcriptional coactivator for genes involved in fatty acid oxidation [46]. QTL for milk protein and casein concentrations [35] and milk yield [47] have been detected at the *LPIN1* locus, which encodes this enzyme, in BTA11, in cattle.

Before they can be assembled into triglycerides, fatty acids first need to be obtained either from the diet or from de novo synthesis. Several genes involved in this latter pathway have also shown QTL for milk-related phenotypes. The rate-limiting step in fatty acid synthesis is the carboxylation of acetyl-CoA to malonyl-CoA, catalysed by the enzyme acetyl-CoA carboxylase (ACC). The alpha form of this enzyme is encoded by the gene acetyl-CoA carboxylase alpha (*ACACA*) on BTA19, and QTL for fatty acid composition [48] and somatic cell score (a proxy phenotype for mastitis) [35] have been mapped to this locus. Another important enzyme in the fatty acid synthesis pathway is fatty acid synthase (FAS), encoded by the gene *FASN* on BTA19, dimers of which are responsible for synthesising the saturated C16 fatty acid palmitic acid from acetyl-CoA and malonyl-CoA [49]. QTL attributed to this gene have been identified for milk fat yield [8], fat concentration [9,50], and fatty acid composition [51] in cattle.

The fatty acid synthase gene will produce only saturated fatty acids. To generate monounsaturated or polyunsaturated fatty acids, desaturase enzymes are required. One cluster of fatty acid desaturase genes on BTA29 includes the two genes fatty acid desaturases 1 and 2 (*FADS1* and *FADS2*); these enzymes are responsible for the final, rate-limiting steps in omega-3 and -6 fatty acid syntheses [52,53]. Variants mapping in cattle to these two genes have been associated with concentrations of a range of polyunsaturated fatty acids in milk [54], and similar associations are also observed in human milk [55,56]. A third desaturase enzyme involved in fatty acid synthesis is stearoyl-CoA desaturase, which is encoded by the gene *SCD* on BTA26 and responsible for oxidising the C16 and C18 saturated fatty acid compounds palmitoyl- and stearoyl-CoA into the monounsaturated compounds palmitoleoyl- and oleoyl-CoA, respectively [57]. This enzyme is also important in regulating metabolism: knockout mice exhibited lower levels of tissue triglycerides and low-density lipoproteins [58], and showed increased insulin signalling and glucose uptake in muscle tissue [59]. In cattle, a QTL assigned to *SCD* has been identified for milk fat yield [8,50].

### 2.3. Hormones and Signalling

Hormones, and the receptors and signalling pathways they activate, are important in most if not all biological functions, and milk production is no exception. For example, knocking out the *SCD* gene discussed in the previous section leads to an increase in tyrosine phosphorylation of the insulin receptor, which has downstream effects on the PI3K-Akt signalling pathway, leading to increased levels of the glucose transporter GLUT4 in the plasma membrane and increased glucose uptake in muscle [59]. These knockout mice also showed lower levels of the hormone leptin, which is involved in regulating energy intake and partitioning. In cattle, leptin (encoded by the gene *LEP* on BTA4) has been associated with both milk yield and feed intake [60]. No milk phenotype QTL have been reported for the leptin receptor (*LEPR* on BTA3), though an association has been reported with body size [61]. In both humans [62] and mice [63], leptin resistance is associated with obesity. Another gene associated with obesity [64] is *FTO*, encoding the enzyme FTO alpha-ketoglutarate-dependent dioxygenase. This enzyme is involved in DNA repair; specifically, it demethylates 3-methylthymidine [65]. It can also demethylate bases in RNA, including 3-methyluracil and 6-methyladenosine [66]. Via this latter mechanism, FTO can inhibit adipogenesis by demethylating cyclin A2 and cyclin-dependent kinase 2 mRNA, reducing the expression of these genes and thereby prolonging the cell cycle [67]. In cattle, variants at the *FTO* locus on BTA18 have been associated with milk fat yield [68].

One hormone of particular importance for lactation is prolactin, a peptide hormone secreted by the anterior pituitary gland. In cattle, this peptide is encoded by the *PRL* gene on BTA23. The importance of this hormone in cattle was underlined by the discovery of a dominant missense mutation that caused, among other phenotypes, a failure to lactate [69]. Prolactin promotes development of the mammary gland during pregnancy, in conjunction with progesterone—generating ductal branching and alveolar buds [70]. Prolactin is detected by cells using prolactin receptor (*PRLR*; BTA20) and acts via the PRLR/JAK2/STAT5 signalling pathway (see Figure 3) to promote mammary gland development and milk protein expression [71], and it induces the expression of the enzyme UDP-glucose pyrophosphorylase 2 (*UGP2*) and the transporter UDP-galactose transporter 2 (*SLC35A2*), thereby promoting the synthesis of lactose [72]. In parallel, prolactin receptor also acts via the PI3K/Akt pathway to downregulate repressors of PRLR/JAK2/STAT5 signalling [73]. Akt1 also upregulates fat synthesis and glucose uptake into the cell for lactose production [74,75]. More recently, it has been shown that Akt signalling induces developing mammary epithelial cells to express prolactin, which in turn acts in an autocrine manner via STAT5 to cause terminal differentiation of the mammary epithelium [71,76]. Given the importance of these pathways, it is not surprising that QTL for milk yield have been widely identified at the *PRLR* locus [35,77,78] and for somatic cell score [78]. Likewise, many studies have reported milk-related QTL at the signal transducer and activator of transcription 5 (*STAT5*) locus on BTA19 (genes *STAT5A* and *STAT5B*) [8,23,24,51,79].

Beyond the leptin and prolactin pathways, other hormones have also been linked to lactation. One example is growth hormone (GH), also known as somatotropin, which comprises part of the same family of protein hormones as prolactin [80]. Growth hormone acts in the lactating animal to partition nutrients and energy towards the mammary gland [81], which is thought to be mediated by increased serum insulin-like growth factor (IGF-1) levels [82]. In the cow, growth hormone is encoded by the gene growth hormone 1 (*GH1*) on BTA19, and its receptor, encoded by growth hormone receptor (*GHR*), maps to BTA20. Both the *GH1* locus [35], and especially the *GHR* locus [9,60,83,84,85], have been associated with milk, fat, and protein yield phenotypes in a range of cattle populations.

Another important family of signalling molecules is the interleukin family of cytokines, a group of primarily immunomodulatory proteins that operate in a paracrine or autocrine manner to affect cell growth and differentiation during immune responses [86]. The gene colony stimulating factor 2 receptor subunit beta (*CSF2RB*) on BTA5 encodes the β-chain of GM-CSF (also known as CSF2) receptor, and also forms a common subunit with the receptors for interleukin-3 (IL-3) and IL-5. QTL for milk, fat, and protein volume; and fat and protein concentrations, have been identified at this locus [8,21,87,88]. *CSF2RB* is suggested as the most likely candidate causative gene, although the neighbouring genes neutrophil cytosolic factor 4 (*NCF4*) [87,88] and thiosulfate sulfurtransferase (*TST*) [9] have also been put forward as candidates. The GM-CSF receptor operates via the JAK/STAT signalling pathway—specifically, JAK2 and STAT5 [86], the same pathway as used by prolactin signalling.

In contrast to STAT5, which is linked to increased milk protein and lactose expression, STAT3 has been identified as a mediator of involution and apoptosis in the mammary gland [89], which is primarily activated by the cytokine leukaemia inhibitory factor (LIF) [90,91]. STAT3 phosphorylation is upregulated within one hour of physical distension of the mammary gland in cattle, ultimately leading to the cessation of milk production when the gland is not emptied [92]. Mice where either *STAT3* or *LIF* is knocked out show delayed involution and reduced levels of apoptosis [90], raising the possibility of improving efficiency in dairy herds, especially those milking once a day, by breeding for animals lacking one or more of these genes. STAT3 is also involved in leptin signalling, interacting with the long form of the leptin receptor [93]. It is difficult to assign QTL unambiguously to the *STAT3* or *STAT5* loci, as *STAT3* maps on BTA19 directly between the two STAT5 genes *STAT5A* and *STAT5B*. Yet another STAT protein, STAT1, is believed to be involved in the development of the mammary gland [94]. A QTL mapping to the *STAT1* locus on BTA2 has been associated with milk, fat, and protein yield [94].

One group of proteins that is upregulated by the JAK/STAT pathway is the suppressors of the cytokine signalling (SOCS) gene family [95]. Proteins in this family downregulate JAK/STAT-mediated signalling. For example, SOCS1 (*SOCS1* on BTA25) expression is induced by prolactin signalling, and in turn binds to JAK2, inhibiting its association with STAT5 and dampening signal transmission [70,96]. SOCS2 (*SOCS2* on BTA5) negatively regulates GH signalling. *SOCS2* knockout mice showed higher body weights and skeletal dimensions than control mice [97]. As SOCS proteins ultimately downregulate milk protein expression via prolactin and growth hormone signalling pathways, there is the potential that knocking out the genes encoding them could improve lactation phenotypes in dairy animals. For example, mice with homozygous *Socs1* knockout genotypes showed enhanced alveolar development [96], and a point mutation in the ovine *Socs2* gene has been associated with higher milk production in dairy sheep, albeit with increased susceptibility to mastitis [98]. In cattle, several different *SOCS* genes have been associated with milk volume, fat, and protein phenotypes [95].

### 2.4. Transporters and Ion Channels

Another important mechanism affecting milk production involves trans-membrane transport and ion channels. All major milk components need to either be produced within the mammary epithelial cells or transported across them from the blood, and in both cases need to cross the apical membrane into the lumen. While some small molecules, such as urea, can simply diffuse across the membrane, in most cases either passive or active transport channels are required.

The volume of water excreted into the milk is driven by osmotic pressure, which is in turn created by exporting lactose and ions across the cell membrane against their concentration gradients. This means that QTL for milk phenotypes frequently map to genes encoding transporters. One important group is the sugar transporters. The gene solute carrier family 37 member 1 (*SLC37A1*) (on BTA1) encodes a phosphate-linked, glucose-6-phosphate antiporter [99], which is responsible for importing glucose into the cell, and QTL for milk volume have been detected at this locus in several populations [21,87,100]. Another glucose transporter, SWEET1, is encoded by the gene *SLC50A1* on BTA3. SWEET1 has been observed in the Golgi in mammary cells and is possibly responsible for importing glucose into the Golgi for lactose synthesis [101]. QTL for lactose concentration [24] and protein concentration [87] have been identified near this gene, though the latter QTL has been assigned to *GBA1*, which encodes the lysosomal protein glucosylceramidase beta 1. Other glucose transporters have also been implicated in milk production, such as GLUT1 (*SLC2A1* on BTA3) [102], the ${\mathrm{Na}}^{+}$/glucose co-transporter SGLT1 (*SLC5A1* on BTA17) [103], and GLUT12 (*SLC2A12* on BTA9) [103].

While the mammary epithelial cells use osmotic pressure to secrete milk, it is important that they maintain their own cell volumes correctly. One mechanism by which they can do this is using voltage-regulated anion channels (VRACs), which help regulate cell volume by exporting Cl${}^{-}$ ions, and small organic anions such as taurine [104,105]. VRACs are comprised of heteromers of leucine-rich repeat containing 8 (LRRC8) proteins A to E, encoded by the genes *LRRC8A*–*LRRC8E*. In cattle, the locus on BTA3 containing *LRRC8B*–*LRRC8D* has been associated with milk lactose concentration [24,79]. *LRRC8C* is suggested as the likely causative gene on the basis of gene-expression data. Interestingly, *LRRC8C* has been associated with adipocyte differentiation (under the name *fad158*) [106] and is also present in the membranes of MFGs [107], suggesting it may also have a role in the storage or export of fat.

As stated above, it is important that water be able to move across the cell membrane to balance osmotic pressure as milk is synthesised. However, the lipid bilayer of the cell membrane is impermeable to water, requiring channels to facilitate the crossing. These channels are provided by aquaporin proteins (AQPs). At least seven aquaporins are expressed in the mammary gland [108,109], where they are believed to play a role in gland development and in transporting water to the lactating gland for milk synthesis and secretion. For example, *AQP1* is expressed in the capillary endothelial cells, and may be involved in oestrogen-mediated angiogenesis in the developing mammary gland [110], and *AQP5* is expressed in mammary epithelial cells, and the protein pores are moved from the cytoplasm to the apical cell membrane under the regulation of prolactin [111], suggesting a role in milk production. Genetic effects mapping to aquaporin genes have been reported, such as a study in sheep [112] that mapped QTL for milk fat and protein concentrations to a window on OAR3 that contains the genes *AQP2*, *AQP5*, and *AQP6*, albeit alongside *LALBA*, which is also a strong candidate causal gene for milk-related traits, as described in the milk protein section above.

Another important class of transporter is the potassium channel, a type of widely expressed ion channel found in the majority of cell types. The majority of these transporters are encoded by genes named *KCN* for the potassium channel, followed by a letter representing the subfamily. A number of different families of potassium channel have been identified. The largest family, the voltage-gated potassium channels ($K_{v}$, reviewed in [113]), responds to voltage changes in the cell’s membrane potential. This family includes the subfamilies *KCNA*, *KCNB*, *KCNC*, *KCND*, *KCNF*, *KCNG*, *KCNH*, *KCNQ*, *KCNS*, and *KCNV* [113]. One channel, $K_{v}3.3$, which is encoded by the gene *KCNC3* on BTA18, has been associated with milk yield in cattle [23]. Another QTL for both milk volume and fat yield maps to the gene *KCNS2*, encoding $K_{v}9.2$, on BTA14 [85]. A third gene in the same family, *KCNH4*, encodes the channel $K_{v}12.3$, and is a potential candidate for a lactose concentration QTL on BTA19 [24], although the QTL also encompass the genes *STAT3*, *STAT5A*, and *STAT5B*, which are also candidates.

A second large family is the potassium inwardly rectifying channel family ($K_{\mathrm{ir}}$, reviewed in [114]), comprising lipid-gated channels that are activated by phosphatidylinositol 4,5-bisphosphate (${\mathrm{PIP}}_{2}$). These transporters correspond to the gene family *KCNJ* [114]. In cattle, a locus on BTA19 has been associated with QTL for fat and protein concentrations [23], lactose concentration [24,79], and milk yield [8]. This locus contains two genes encoding $K_{\mathrm{ir}}$ channels: *KCNJ2* ($K_{\mathrm{ir}}2.1$) and *KCNJ16* ($K_{\mathrm{ir}}5.1$), and both genes have been proposed as candidate causatives underlying the QTL.

A third, smaller family of potassium channels is the two-pore-domain potassium channels ($K_{2P}$), known as leak channels [115], corresponding to the *KCNK* family. A QTL for milk fat, protein, and volume has been identified on BTA26 at the locus of the *KCNK18* gene, encoding the potassium channel $K_{2P}18.1$ [85]. The fourth family is the calcium- and sodium-activated potassium channels [116], of which the most well-known member is $K_{\mathrm{Ca}}1.1$, also known as BK. The activity of $K_{\mathrm{Ca}}1.1$ is modulated by auxiliary β and γ subunits [117,118], including γ1, encoded by the gene *LRRC26* on BTA11, where a QTL for fat concentration has been detected [23].

The channels discussed so far are limited to moving solutes along an existing concentration gradient. Another class of transporter requires energy in the form of ATP to concentrate solutes to establish a gradient or membrane potential. Many of these belong to the ATP-binding cassette family (ABC). One important example from this family is ATP binding cassette subfamily G member 2 (*ABCG2*). Initially identified as a xenobiotic drug transporter [119], ABCG2 also functions as a transporter of riboflavin into the milk [120] and a urate transporter in the kidneys [121]. QTL for several milk phenotypes have been mapped to the *ABCG2* locus on BTA6, including milk yield [8,21,85,122], fat and protein concentration [9,79,122], lactose concentration [24,79], and $\alpha_{S1}$-CN concentration [100]. The causative variant at this QTL is generally believed to be a tyrosine-to-serine substitution (Y581S), identified by Cohen-Zinder et al. [122]; however, the adjacent gene *SPP1*, encoding the protein osteopontin (involved in bone remodelling), has also been proposed as a candidate [123,124]. A second member of the ABC family is SUR2 (encoded by *ABCC9* on BTA5), which forms a component of the ATP-sensitive potassium channel $K_{\mathrm{ATP}}$, alongside SUR1 (*ABCC8*) and the inward rectifying channels $K_{\mathrm{ir}}6.1$ (*KCNJ8*) and $K_{\mathrm{ir}}6.2$ (*KCNJ11*) [125]. The exact composition varies by tissue [126]. The channel is inhibited by ATP and activated by MgADP. The $K_{\mathrm{ATP}}$ transporter is important for glucose-level sensing to control insulin release in pancreatic beta cells [127]: at low glucose levels, ATP levels are low and ADP levels are elevated, so the channel is open; and at high glucose levels, ATP closes the channel. This polarises the plasma membrane, a change that is detected by voltage-gated calcium channels, which then open, causing calcium to enter the cell and trigger the release of insulin. In cattle, a QTL for milk fat yield has been identified at the *ABCC9* locus [23,84]. A third ATP-binding transporter is the calcium transporter SERCA2, encoded by the gene ATPase sarcoplasmic/endoplasmic reticulum ${\mathrm{Ca}}^{2+}$ transporting 2 (*ATP2A2*) on BTA17. This transporter pumps ${\mathrm{Ca}}^{2+}$ from the cytosol into the endoplasmic reticulum, whence it can be exported into milk [128]. A QTL at this locus has been detected for milk and protein yield, and for milk calcium [37].

Another ion transporter, showing widely reported associations with milk phenotypes, is inorganic pyrophosphate transport regulator, encoded by the gene *ANKH* on BTA20. The ANKH transporter controls extracellular mineralisation by regulating the levels of inorganic pyrophosphate in the extracellular matrix. In humans and mice, mutations in this transporter have been associated with arthritis and bone growth disorders linked to tissue calcification [129,130]. This transporter has previously been assumed to be a pyrophosphate transporter; however, recent work has shown that the channel in fact transports ATP, and that the production of extracellular pyrophosphate from ATP requires the enzyme ectonucleotide pyrophosphatase/phosphodiesterase 1 (ENPP1) [131]. In cattle, QTL mapping to the *ANKH* locus have been identified for milk yield [8,21], milk lactose concentration [24,79], and α-lactalbumin concentration [100]. Another recent study [132] discovered that the ANKH protein cycles between the plasma membrane and trans-Golgi network using clathrin-coated vesicles mediated by clathrin adaptors AP1 and AP2. The phosphatidylinositol binding clathrin assembly protein also interacts with AP2 [133], binding to the signalling molecule phosphatidylinositol and recruiting the AP2 complex to form clathrin-coated pits. This protein is encoded by the gene *PICALM* on BTA29 and is associated with QTL for $\alpha_{S1}$-CN [100] and lactose concentration [24,79].

## 3. Identifying Candidate Causative Genes

The previous section may have given the impression that identifying the causative genes, or even variants underlying QTL, is easy. However, in many cases, there will be no obvious candidate genes in the QTL, possibly because the causative variant sits in a long-range regulatory element for a distant gene. In other cases, there may be many potential candidate genes, and different selection methods may highlight different candidates. For example, the window from 42.2 to 42.5 Mbp on BTA19 envelopes several candidate genes for milk phenotypes. A 2015 study by Raven et al. [87] highlighted the genes GH3-domain containing (*GHDC*), *STAT5A*, and *STAT5B* on the basis of differential expression and enrichment of significant variants, and proposed *STAT5A* as causative on the basis of knockout studies in mice. In contrast, later work by our group [24,79] used eQTL data and missense variants in strong LD with the top QTL variant to highlight the genes DExH-box helicase 58 (*DHX58*), *GHDC*, lysine acetyltransferase 2A (*KAT2A*), *KCNH4*, *STAT5A*, and *STAT5B*. The two STAT5 genes were again proposed as the most likely candidates. As discussed above, potassium channels, such as that encoded by *KCNH4*, have been associated with milk at a number of loci, and the histone deacetylase and transcription activator *KAT2A2* is also a possible candidate, based on the high levels of gene expression required in the lactating mammary gland. It is possible that multiple QTL are segregated at this locus, and therefore, that more than one gene is causative.

Another region with two strong candidate causal genes maps to between 15.3 and 15.6 Mbp on BTA3. At this locus, Raven et al. [87] identified a QTL for milk protein concentration and proposed the epithelial mucin gene *MUC1* as the best candidate. The mucin 1 protein coded by this gene is considered a “metabolic master regulator” [134], regulating tyrosine-kinase signalling and the expression of metabolic genes. Work by our group [24,79] has identified a QTL for milk lactose concentration at the same locus, and proposed the gene *SLC50A1*, encoding a sugar transporter. Again, it is possible that these are two separate QTL, and that both genes are causative for the corresponding phenotypes; however, another possibility is that the QTL is pleiotropic, and only one QTL is present controlling both phenotypes. Distinguishing between these two possibilities is difficult or impossible using purely statistical or bioinformatic means, and additional experiments are likely to be required.

### 3.1. Molecular Phenotypes

Widely used phenotypes, such as fat and protein, effectively aggregate signals from a large number of milk components: individual fatty acids and other lipids for fat, and a number of casein and whey proteins for protein, for example. It is likely that these different components are not all under the same genetic regulation, and some correlations may even be negative. For example, Stoop et al. [135] observed genetic correlations of as low as −0.84 between milk fatty acid measures (C14:0 and C16:0), and correlations over 0.9. This suggests that using finer composition measures, such as individual fatty acids or proteins, should give cleaner, and possibly a larger number of genetic signals compared to complex phenotypes such as fat or protein. This approach has been successfully applied to use milk minerals and individual protein measurements to identify novel QTL and to highlight *SLC37A1* as involved in lactation [37]. These phenotypes can be considered “molecular”, as measurements for a single molecule, for example, using gas or liquid chromatography with mass spectroscopy, provide the phenotype.

Molecular phenotypes in milk are often measured using Fourier-transform mid-infrared spectroscopy, as this method is commonly used for commercial herd testing and is cheaper to perform on a large scale than spectroscopy. This method uses the absorbance of around 900 different frequencies of infrared light (known as wavenumbers), then uses these values in a model to predict the phenotypes of interest. In commercial herd testing, the predicted phenotypes are typically fat, protein, and lactose concentrations; however, models have been developed for a range of other phenotypes, such as individual fatty acids [136] and proteins [137], and phenotypes more remote from milk, such as methane production [138] and fertility [139]. However, because FT-MIR phenotypes are predicted rather than measured, their utility for QTL discovery can be variable. For example, our recent work [140] showed that both HPLC measured and FT-MIR predicted phenotypes could detect significant *cis*-QTL for α-casein, κ-casein, and β-lactoglobulin. However, FT-MIR was unable to identify the highly significant *cis*-QTL for lactoferrin, which was identified using HPLC, and identified a QTL at the locus of the fat synthesis gene *DGAT1* for α-casein. The latter may be due to signal crossover from fat, which is highly correlated with protein in milk. This form of crossover, if present, would complicate the task of identifying pleiotropy.

In addition to predicting concentrations of milk components, it is also possible to use the individual wavenumber data themselves as phenotypes [79]: different wavenumbers represent different chemical bonds that absorb MIR light at the corresponding frequencies, so wavenumber phenotypes effectively measure the concentrations in milk of compounds that contain specific chemical bonds. Wavenumber phenotypes have been shown to detect stronger and more numerous genetic signals compared to the predicted phenotypes such as fat and protein [79].

Examining the expression level of each gene in a relevant tissue sample provides an alternative to measuring or estimating protein concentrations. In milk, for example, expression levels of the genes encoding milk proteins in the lactating mammary epithelial tissue can give proxy measurements for milk proteins. Traditionally, this could be done using techniques such as qPCR or expression microarrays. More recently, RNA sequencing (RNA-seq) being used to sequence and count (relatively) mRNA molecules has become common. As RNA-seq captures data for every expressed gene, it is possible to search for expression QTL (eQTL) and allele-specific expression (ASE) for every gene, including those not expressed in the milk, such as fat synthesis enzymes and hormone receptors. The presence of a QTL for a given gene can provide evidence on the causality (or otherwise) of the gene at an overlapping QTL: when the QTL is caused by an underlying eQTL, we expect that the causal variant or variants will be strongly associated with both the QTL and eQTL, and that associations for other variants will decay away in proportion to linkage disequilibrium with the causal variants. This pattern can be identified by examining the correlations between variants regarding the QTL and eQTL variants’ effects [141], or p values (on a logarithmic scale) [24,45], or correlations between local genomic estimated breeding values (GEBVs) and gene expression [142]. Other methods used to associate eQTL with GWAS results include transcriptome-wide association scans (TWAS) [143], Mendelian randomisation [144,145], and Bayesian colocalisation methods [142,146].

Molecular phenotypes and their QTL have assisted in highlighting candidate causative genes in cases where no obvious candidates existed. For example, the gene *MGST1* on BTA5 encodes microsomal glutathione S-transferase 1, which belongs to a family of detoxification enzymes [147]. It has no obvious role in milk production. Nevertheless, QTL for milk traits mapping to this locus have been reported in many different studies [21,87,100,148,149]. The neighbouring gene *EPS8* (epidermal growth factor receptor kinase substrate 8) is sometimes proposed as a candidate. However, gene-expression data from lactating bovine mammary tissue have shown that the milk QTL at this locus co-segregate with an eQTL for *MGST1*, whereas *EPS8* is barely expressed in this tissue. Similarly, a QTL for milk-fat percentage on the distal end of BTA11, has been linked to the gene encoding the ABO blood group, which is also named *ABO* (alpha 1-3-N-acetylgalactosaminyltransferase and alpha 1-3-galactosyltransferase), using RNA-seq data to show that a splice donor site mutation causes aberrant splicing of the *ABO* transcript, in turn causing an eQTL that co-segregates with the fat percentage QTL [79].

### 3.2. Chromatin Structure Phenotypes

Chromatin is the name given to the compound comprising DNA wound around nucleosomes, which are themselves composed of histone proteins. The three-dimensional folded structure of chromatin can have an important impact on gene expression. On a proximal scale, the structure of the chromatin region surrounding a gene can be in open or closed configurations. The former provides access for transcription factors and RNA polymerase components to reach the promoter region and trigger gene expression, and the latter, closed, configuration blocks expression. Variants sitting within an open chromatin region in the appropriate cell type are more likely to have a regulatory impact on gene expression [141], and therefore, are more likely to present good candidate causal variants for a given trait, compared to other non-coding variants.

One method commonly used to study chromatin state is chromatin immunoprecipitation (ChIP) [150], which can be used to identify transcription-factor binding sites or histone modifications. Histone proteins feature long tails, and a wide variety of post-translational modifications can be made to these tails to alter the chromatin state. For example, the modifications H3K4me2 (histone 3, lysine 4 dimethylation), H3K4Me3, and K3K27ac (lysine 27 acetylation) are associated with open chromatin in active promoters and enhancers; and closed, repressed regions can be marked by H3K9me2 or H3K27me3 [151]. More recent versions of ChIP, using DNA sequencing techniques to target the whole genome in a single experiment, include ChIP-seq [152,153] and CUT&RUN-seq [154]. Recent work [141] has identified *cis*-QTL for ChIP-seq phenotypes; and demonstrated that these QTL frequently exhibit strong correlation with nearby eQTL, and that eQTL tag variants are significantly enriched within ChIP-seq-identified open chromatin windows; this illustrates the utility of this type of data for understanding regulatory variation in the genome.

Open chromatin is accessible to transcription factors and other proteins required for gene expression. However, it is also more accessible than other regions to nuclease enzymes, and this fact is used in another approach to identifying open chromatin regions. One commonly used nuclease is DNase 1, which can be used in conjunction with high-throughput sequencing via DNase-seq [155]. The hypersensitivity sites identified using this technique highlight the positions of regulatory elements such as promoters, silencers, and enhancers. A similar technique, using the enzyme micrococcal nuclease, is called MNase-seq [156,157], and identifies the positions of nucleosomes. As nucleosomes are scarcer in open chromatin, these data give an inverse signal to hypersensitivity sites identified by DNase-seq. A more recent technique called the assay for transposase accessible chromatin (ATAC-seq) [158] uses a modified hyperactive transposase enzyme Tn5 to fragment and load sequencing adaptors to open chromatin regions. ATAC-seq typically requires fewer cells and less time to perform compared to DNase-seq, although single-cell protocols have now been developed for both methods [159,160].

On a larger scale, the 3D folded structure allows distal enhancer and silencer elements to enter close contact with the gene to perform their respective functions on gene expression [161]. Identifying this structure allows linkages between genes and distal regulatory elements to be identified. These sorts of long-range interactions can be studied using chromosome conformation capture (3C) [162], and related methods, such as 3C-on-chip (4C) [163] and 3C carbon copy (5C) [164]. These older techniques can study only small parts of the genome. A more recent technique, Hi-C [165], could originally study the structure of the whole genome only at low resolution (≈1Mbp), but later refinements of the method [166,167] have allowed for improved resolution and signal-to-noise ratios.

### 3.3. From Candidate Genes to Causative Variants

The results of GWAS and similar experiments are typically genomic intervals containing several associated variants that are in strong LD with one another. Although candidate causal genes may be selected using known pathways, gene-expression data, or chromatin structure, as discussed above, for some applications it is useful or necessary to know the causal variant underlying the QTL. These applications include improving the accuracy of genomic selection, performing genetic testing, or creating gene-edited animals [168]. However, identifying the causal variant using purely statistical or bioinformatics approaches is typically not possible, as variants in strong LD cannot be distinguished from one other statistically.

In some cases, variant annotation (using tools such as SnpEff [169] or Ensembl’s Variant Effect Predictor [170]) can highlight missense or nonsense mutations that will make stronger candidates [41], but many QTL are driven by regulatory effects rather than coding ones. This means that other tools will be needed to resolve QTL with underlying regulatory effects. One commonly used technique for discovering *cis*-regulatory elements (CREs) is transcription factor binding site (TFBS) prediction. Several tools have been developed for this, generally using information from databases containing binding site motifs for large numbers of transcription factors, such as TRANSFAC [171] and JASPAR [172]. A second approach to investigating regulatory effects is to use a reporter assay, where the effects of putative promoter variants can be tested against the expression of a reporter gene such as GFP. This method has recently been scaled up to test thousands of variants simultaneously using a massively parallel reporter assay (MPRA) [173]. A third method is to use CRISPR-Cas9 or other gene-editing technologies to test putative regulatory sequences, either by deleting them using NHEJ, or by editing in alleles of interest using HDR and then observing the resulting effect on gene expression, often using single-cell RNA-seq. Methods incorporating CRISPR include Perturb-seq [174], CROP-seq [175], and HCR-Flowfish [176].

## 4. Conclusions

Over the last twenty years since the seminal work of Grisart et al. [42] showed that the *DGAT1* gene underlies the large milk QTL on BTA14, a large number of milk-related QTL have been identified in cattle, and candidate causative genes have been proposed for many of them. In the coming years, we can expect that sequence-resolution data sets will continue both to grow and diversify to include additional breeds from around the world. These larger, more diverse data sets will likely empower the discovery of many novel QTL. Although proving the causality of genes and variants underlying these QTL will likely remain difficult, the use of molecular phenotypes, massively parallel reporter assays, and CRISPR will extend the list of proven causatives. This new information should provide greater insights into the genes and pathways underlying the initiation and maintenance of lactation in mammals. Knowledge of causative genes and variants may also improve the accuracy of genomic selection in animal breeding, which would accelerate genetic gain and improve on-farm productivity. Causal variants will also provide for gene editing to rapidly spread beneficial variants through the population, should future regulatory environments allow it.

## Figures and Tables

**Figure 1 animals-13-00911-f001:**
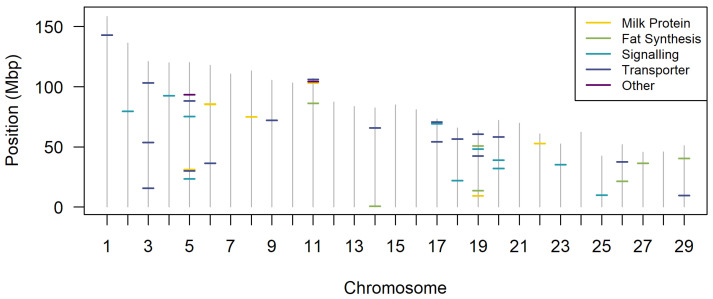
Locations on the 29 bovine autosomes of the QTL discussed in this paper. Colors indicate which of the sections the QTL is discussed under. The “other” category represents the two genes, *MGST1* and *ABO*. Positions are based on the ARS-UCD1.2 reference genome.

**Figure 2 animals-13-00911-f002:**
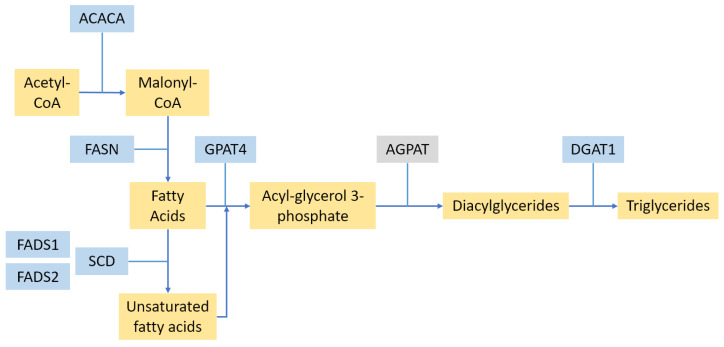
An overview of fatty acid and triglyceride synthesis, highlighting the enzymes discussed in Section 2.2. Enzymes are shown in blue, and other products in yellow. The AGPAT enzyme group is shown in grey, as the only AGPAT gene exhibiting a significant QTL (*AGPAT6*) has been renamed to *GPAT4*.

**Figure 3 animals-13-00911-f003:**
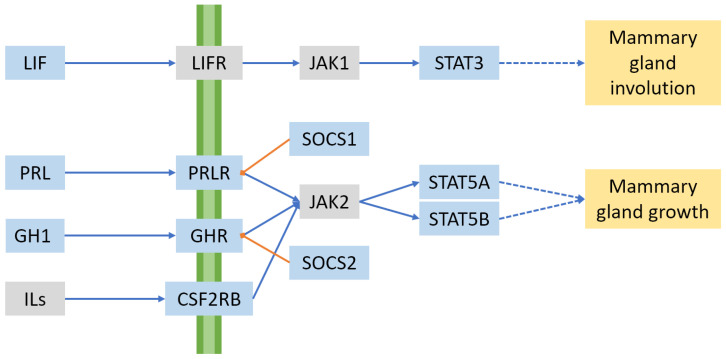
JAK/STAT signalling pathways with effects on mammary gland growth or involution. Signalling proteins, receptors, and transducers with known milk QTL are shown in blue, and selected other components are shown in grey. Blue arrows represent activation, and orange arrows indicate suppression. The green band represents the plasma membrane.

**Table 1 animals-13-00911-t001:** Top milk yield and composition QTL counts for cattle by trait. Data from CattleQTLdb release 49 (28 December 2022) [3].

Trait	Number of QTL Variants
Milk fat percentage	11,911
Milk protein percentage	9958
Milk fat yield	9255
Milk yield	7383
Milk C14 index	4847
Milk kappa-casein percentage	4275

## Data Availability

No new data were created or analysed in this study. Data sharing is not applicable to this article.

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
