# Peer review of "Using QTL to Identify Genes and Pathways Underlying the Regulation and Production of Milk Components in Cattle"

_animals, 2023, doi:10.3390/ani13050911_

Round 1

Reviewer 1 Report

This review describes the QTL identified in milk, showing that a large number of milk-related QTL have been identified in milk, and describes many of these candidate associated genes and elaborates on the role of these genes with milk synthesis. The genes of these QTL have multiple functions. Some, such as prolactin and growth hormone, and their associated signaling pathways, are involved in breast enlargement: mammary gland development during puberty and pregnancy. Other pathways, such as fat and protein synthesis, and ion channels, are directly involved in milk production. However, there are still many shortcomings, as described below.

1. The abstract states that other ruminants are described, please describe exactly which ones.

2. The conclusion section does not have a clear outlook on the results and the future expectations of genes controlling quantitative traits in improving the quality of milk.

3. The article describes protein synthesis and milk fat synthesis in milk, but does not describe free lactose synthesis in milk. it is suggested to add a new paragraph to describe the functional role of QTL and their related candidate genes in free lactose synthesis in milk.

4. The article in 2.1, 2.2, 2.3, 2.4 suggests drawing pictures showing the mechanisms of regulation.

5. Please give the full names of all genes appearing for the first time in the chapter, e.g. line45: "CSN1S1", line46: "CSN1S2", etc.

Author Response

> The abstract states that other ruminants are described, please describe exactly which ones.

The other ruminant species mentioned occasionally in the text was sheep, and the text has been changed to reflect this.

> The conclusion section does not have a clear outlook on the results and the future expectations of genes controlling quantitative traits in improving the quality of milk.

I have expanded the conclusion section to state that knowledge of causal genes and variants may improve the rate of genetic gain via improvements in genomic selection, and mentioned gene editing as a potential future use.

> The article describes protein synthesis and milk fat synthesis in milk, but does not describe free lactose synthesis in milk. it is suggested to add a new paragraph to describe the functional role of QTL and their related candidate genes in free lactose synthesis in milk.

> The article in 2.1, 2.2, 2.3, 2.4 suggests drawing pictures showing the mechanisms of regulation.

An excellent idea. I have added new figures for fat synthesis pathways (Fig 2, section 2.2) and for JAK/STAT signalling pathways (Fig 3, section 2.3). I have also added a new figure (Fig 1, section 2) providing an overall view of where in the cattle genome all the QTL mentioned in the paper can be found.

> Please give the full names of all genes appearing for the first time in the chapter, e.g. line45: "CSN1S1", line46: "CSN1S2", etc.

I have added the full gene names where these differ from the protein names already included. The exception to this is when a large number of gene name are similar, such as the many solute carriers (SLC...) or potassium channels (KCN...), when the name is only expanded for the first example.

Reviewer 2 Report

The review is interesting and  well written.

I have only a request.: the author should explain why, mainly in the first part of the review, he refers in knowledge in the human species and therefore to the production of human milk.

Lines 47 and 50. Change Miscelle in MICELLE/

Author Response

> I have only a request.: the author should explain why, mainly in the first part of the review, he refers in knowledge in the human species and therefore to the production of human milk.

This was done when the composition of cattle milk differed in a major way from human milk, in order to make the review useful to a wider range of readers. The introduction text has been altered to make this clearer.

> Lines 47 and 50. Change Miscelle in MICELLE/

Done. Thanks for finding and pointing out the error

Reviewer 3 Report

The manuscript provides a comprehensive view of QTL/genes for mink production and components. I have some minor suggestions:

A simple summary is required for the manuscript.

Line 23-32: The author might add some sentences about the methods for QTL detections and candidate gene identifications.

Line 34: The author might check the AnimalQTL database and provide the numbers of QTL for mink-related traits, or some updated information about state-of-the-art in this aspect.

Line 379: How about the non-coding RNAs, the authors might shortly summarize them as they are important regulators of mink production and components. 

Author Response

> Line 23-32: The author might add some sentences about the methods for QTL detections and candidate gene identifications.

I have added a sentence mentioning several techniques used for QTL discovery. I have also added a sentence to the end of the paragraph referring the reader to Section 3 where methods of candidate gene identification are discussed

> Line 34: The author might check the AnimalQTL database and provide the numbers of QTL for mink-related traits, or some updated information about state-of-the-art in this aspect.

This is a great idea, and I have added a table showing the numbers of QTL identified for cattle in the AnimalQTLdb for several milk-related traits.

> Line 379: How about the non-coding RNAs, the authors might shortly summarize them as they are important regulators of mink production and components. 

Another interesting idea, and a good topic to review, but unfortunately I am not knowledgeable enough about miRNA or other regulatory ncRNAs to usefully review the topic myself.

Round 2

Reviewer 1 Report

I didn't find the comment for Figure 1 in the article.